

# Laying the foundations for selective-fish guidance using electricity: multi-species response to pulsed direct currents

Mhairi Miller, Suleiman M. Sharkh and Paul S. Kemp

University of Southampton, Southampton, United Kingdom

## ABSTRACT

To develop effective technology that employs electric fields to simultaneously guide valued freshwater fish whilst limiting the range expansion of undesirable invasive species, there is a need to quantify the electrosensitivity of multiple families. This experimental study quantified the electrosensitivity of two carp species that, in UK, are invasive (grass carp, *Ctenopharyngodon idella*, and common carp, *Cyprinus carpio*) and compared the values with those previously obtained for adult European eel (*Anguilla anguilla*), a species of conservation concern in Europe. Electric field strengths (V/cm) required to elicit physiological responses (*twitch*, *loss of orientation* and *tetany*) were identified across four pulsed direct current (PDC) electric waveforms (single pulse-2 Hz, double pulse-2 Hz, single pulse-3 Hz and double pulse-3 Hz). Grass carp were sensitive to differences in waveform with *tetany* exhibited at lower field strengths in the single pulse-2 Hz treatment. Both cyprinid species responded similarly and were less sensitive to PDC than adult European eel, although *loss of orientation* occurred at lower field strengths for grass than common carp in the single pulse-3 Hz waveform treatment. This variation in electrosensitivity, likely due to differences in body length, indicates potential for electric fields to selectively guide fish in areas where invasive and native species occur in sympatry.

## INTRODUCTION

With the global human population predicted to continue to grow, at least during the first half of the current century (*Adam, 2021*), fresh waters will be increasingly exploited, *e.g.*, to generate energy and produce food. This will require the construction of large dams to supplement those that have already impacted nearly two-thirds of the world's longest rivers (>1,000 km) (*Grill et al., 2019*) in the drive to further economic development (*Shi et al., 2019*). The rapid growth in exploitation of freshwater resources will cause further environmental and social-economic shocks associated with the modification and degradation of ecosystems and the services they provide (*Kemp et al., 2022*), *e.g.*, through the disruption of fluvial connectivity and fragmentation of habitat. From the perspective of freshwater fisheries, dams impede movements between critical habitats of fish that might also be injured or killed if they enter intakes (*e.g.*, to hydropower turbines or irrigation

Corresponding author
Mhairi Miller, m.miller@soton.ac.uk

systems) (*Kemp, 2015*). The development of effective strategies to mitigate the negative impacts of river infrastructure is critical if environmental degradation is to be halted or reversed as society strives to meet sustainability targets (*e.g.*, UN Sustainable Development Goals; *United Nations, 2015*).

As part of a programme to advance the sustainability of river engineering, fish passes and physical and mechanical screens can be built and installed at impoundments to preserve or restore migration routes and thus improve habitat connectivity (*Clay, 1995*) and prevent fish entering associated water intakes (*Kemp, 2015*). Behavioural stimuli such as acoustics (*Jesus et al., 2019*), light (*Ford et al., 2018*), bubbles (*Flores Martin et al., 2021*) and electric fields (*Parasiewicz et al., 2016*) have been tested as an alternative to (*e.g.*, *Noatch & Suski, 2012*), or to enhance the effectiveness of (*e.g.*, *Deleau et al., 2020*; *Mussen & Cech Jr, 2019*), physical and mechanical screens designed to block passage and/or guide fish away from dangerous areas.

Unfortunately, current mitigation strategies are not always fully effective, thus only providing partial solutions, referred to as "half-way" technologies by some (*Brown et al., 2013*). Furthermore, the mitigation can itself be damaging and have unforeseen consequences that are often overlooked or underappreciated (*Mclaughlin et al., 2013*). For example, fish passes that partially reconnect habitat critical for the completion of the life-cycle of desirable native species, such as those with high commercial, cultural, and conservation significance, may also facilitate range expansion of Aquatic Invasive Species (AIS) (*Kerr et al., 2021*). The introduction of AIS can have large negative consequences for recipient ecosystems, acting through predation (*e.g.*, *Weyl & Lewis, 2006*; largemouth bass, *Micropterus salmoides*), parasitism (*e.g.*, *Patrick, Sutton & Swink, 2009*; sea lamprey, *Petromyzon marinus*), resource competition (*e.g.*, *Baker & Levinton, 2003*; zebra mussel, *Dreissena polymorpha*), habitat modification (*e.g.*, *Brown & Moyle, 1991*; Sacramento squawfish, *Ptychocheilus grandis*), hybridisation (*e.g.*, *Ludwig et al., 2009*; Siberian sturgeons, *Acipenser baerii*) and disease transmission (*e.g.*, *Alderman, Holdich & Reeve, 1990*; signal crayfish, *Pacifastacus leniusculus*), as well as causing substantial economic impacts (*Pimentel et al., 2000*). Hence, trade-offs arise when mitigation strategies employed to benefit native species conflict with management decisions to control the spread of AIS (*Mclaughlin et al., 2013*), resulting in a "connectivity conundrum" (*Zielinski et al., 2020*). Consequently, there is a need to enhance conservation efforts that benefit native species, while reducing the risks posed by AIS. Developing selective environmental impact mitigation technologies could reduce the tensions between AIS control and native fish passage objectives (*Rahel & McLaughlin, 2018*).

Previous research to develop environmental impact mitigation technology and AIS control based on selecting specific traits exhibited by the target species has tended to focus on the use of physical structures to facilitate passage rather than guidance. Perhaps some of the earliest examples relate to sea lamprey control in the Laurentian Great Lakes (*Zielinski et al., 2019*). Multiple barrier designs, such as fixed-, seasonal- and adjustable-crest weirs and velocity barriers have been used to limit the movements of invasive sea lamprey during peak migration (*McLaughlin et al., 2007*), while allowing desirable families (such as the salmonids) to gain access to spawning streams. In another case, selective fish passage has

been developed to conserve native Pacific lamprey (*Entosphenus tridentatus*) in the Pacific Northwest of the United States through the design and installation of lamprey passes that compensate for ineffective technical fishways designed for salmonids (*Moser et al., 2011*).

Behavioural deterrents designed to selectively block and/or guide fish away from dangerous areas to more preferred routes provide a management option to protect desirable species while deterring AIS (*Noatch & Suski, 2012*). Early examples of the use of low-voltage electric fields for sea lamprey control in the Great Lakes date back to the 1950s (*Applegate, Smith & Nielsen, 1952*; *Erkkila, Smith & McLain, 1956*; *McLain, 1957*). A more recent example of the use of electric barriers relates to that activated in the Chicago Sanitary and Ship Canal in 2002, the world's largest such device, designed to prevent the interbasin transfer of AIS (in particular bighead carp, *Hypophthalmichthys nobilis*, and silver carp, *Hypophthalmichthys molitrix*), primarily from the Mississippi to the Great Lake catchments (*Parker et al., 2015*). Unfortunately, electric barrier and guidance devices too can have negative unintended environmental impacts, dating back to the early studies in which alternating current was used to block sea lamprey but resulted in excessive mortality of non-target species (*Applegate, Smith & Nielsen, 1952*; *Erkkila, Smith & McLain, 1956*). Despite improvements in understanding and design, this risk remains, as illustrated during the evaluation of the effectiveness of a portable seasonal electric barrier installed in a tributary of the Great Lakes for sea lamprey control, but that also blocked and killed hundreds of non-target fish (*Johnson et al., 2021*).

Even when electric barriers work, their efficiencies can be variable. Some demonstrate close to 100% barrier effectiveness (*e.g.*, *Swink, 1999* for invasive sea lamprey; *Sparks et al., 2011* for common carp, *Cyprinus carpio*); whereas others show greater variability in efficiency if designed for guidance (*e.g.*, *Gosset & Travade, 1999*, 5–28% for native Atlantic salmon smolts *Salmo salar*; *Johnson & Miehls, 2013*, 84% for sea lamprey). To advance selective electric barriers and fish guidance systems that respectively facilitate the control of AIS and conservation of native species, there is a need to develop fundamental knowledge of sensory capabilities (*e.g.*, electrosensitivities) and behavioural response from a multi-species/fish community perspective. This includes quantifying intra- and interspecific response to electric fields exhibiting different characteristics, such as field strength, pulse frequency and width, that are known to influence the behaviour and physical condition (*e.g.*, *Larson, Meyer & High, 2014*; *Layhee et al., 2016*) of different life-stages (*e.g.*, *Nutile, Amberg & Goforth, 2012* for zebrafish, *Danio rerio*, embryos; *Miller et al., 2021*; *Miller, Sharkh & Kemp, 2022* for adult and juvenile European eel).

To inform the design of selective electric barriers, this experimental study used model species of high conservation concern (European eel) and those that are invasive (grass carp, *Ctenopharyngodon idella*, and common carp, *Cyprinus carpio*). The European eel is critically endangered throughout its range, having suffered declines in recruitment of 90–99% since the 1980s (*ICES, 2017*). Carp species represent a considerable risk in regions where they are non-native, *e.g.*, through competition, habitat alteration, and spread of disease (*Manchester & Bullock, 2000*). Both species co-occur in many European rivers (*e.g.*, *Cooper & Wheatley, 1981*; *Lehtonen, 2002*; *Nunn et al., 2007*; *Nunn et al., 2011*; *Van der Veer & Nentwig, 2015*). We quantified and compared threshold field strengths (V/cm) for three physiological

responses (*twitch, loss of orientation and tetany*). The influence of the following parameters on threshold field strengths for the three physiological responses were assessed: (1) Pulsed Direct Current (PDC) waveforms differing with respect to pulse width and frequency (Objective 1); (2) species (grass and common carp) (Objective 2); and (3) family (cyprinid and anguillid) (Objective 3).

## MATERIALS & METHODS

### Ethical note

All methods were followed in accordance with the UK national regulatory Animal Welfare Ethical Review Body's (AWERB) guidelines and regulations. This study was approved by the University of Southampton's Ethics and Research Governance Office (Ethics ID 45107 & 30639) and reported in accordance with ARRIVE guidelines.

### Experimental set-up

Experiments were conducted at the International Centre for Ecohydraulics Research (ICER) facility, University of Southampton, UK, using a clear glass (10 mm thick) walled rectangular tank (1.5 m long × 0.60 m wide × 0.23 m deep) (Fig. 1) (see *Miller et al., 2021*). Two aluminium plate electrodes (0.5 m wide × 0.35 m high × 2 mm thick) were placed at either end of the tank 1.42 m apart and an electrically insulating mesh screen (0.56 m wide × 0.23 m high × 2 cm deep, mesh diameter = 1 mm) was placed in front of each to prevent contact with the fish. The electrodes were connected to an ETS ABP-2 backpack electrofisher (ETS Electrofishing Systems LLC) modified as a pulse generator (200 W average output; 600 V/10 A maximum peak outputs), powered by a 12 V DC battery.

The electric field was mapped using a potential probe consisting of two-point conductors 27 mm apart connected to an oscilloscope (Gwinstek GDS-1052-U) *via* a differential probe module (Probemaster Model 4232). Measurements were taken in a grid at a spacing of 10 cm in the *x* and *y* direction and at two depths (5 and 10 cm depth from the water surface) to record peak-to-peak voltage. Electric field maps were generated for all output voltages and waveforms. Electric field strength was uniform across the tank and proportional to output voltage (*i.e.,* for 7 V output voltage with 142 cm between electrodes: $\frac{7}{142} = 0.05$ V/cm) (see *Miller et al., 2021*). Ambient water conductivity during mapping was 630 µS. cm$^{-1}$.

Four CCTV cameras (Swann 1080p; 1,920 ×1,080 pixel resolution) were used to monitor and record fish behaviour: two overhead (1 m directly above the tank rim), and two side-facing (34 to 39 cm perpendicular to the tank side). Two infrared lights (780–850 nm wavelength) were placed above the tank (70 cm from each camera) to provide additional illumination during periods of darkness.

Water (conditioned tap water) depth was maintained at 15 cm and obtained from the holding tank in which the experimental fish used that day were housed.

### Fish husbandry

Forty grass carp and forty-five common carp were obtained from a supplier (Aquatics to your Door Ltd., 10 August 2018) and local hatchery (Hampshire Carp Hatcheries, 16

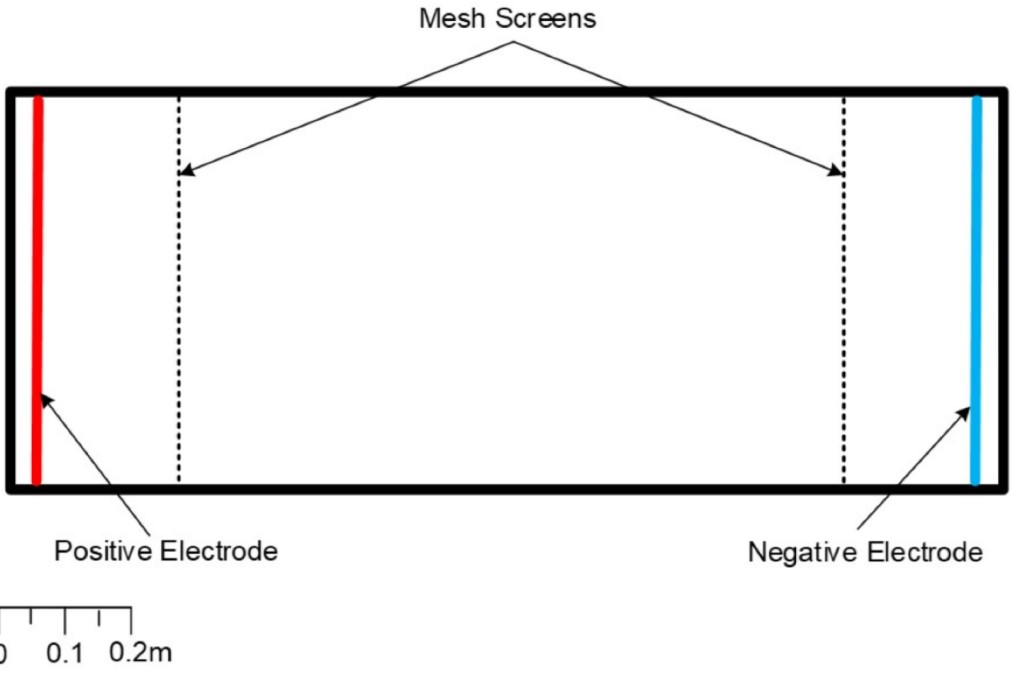

**Figure 1** **Plan view of the glass tank used to test electrosensitivity of cyprinids and adult European eel (adapted from *Miller et al., 2021*) to PDC electric fields.** Two aluminium electrodes were placed at either end of the tank and connected to a pulse generator to provide the electric field.

August 2018), respectively. Forty European eel were obtained from commercial fisherman using fyke nets on 26 October 2017 (see *Miller et al., 2021*). The fish were split evenly over four 3000 litre outdoor holding tanks with sections of plastic piping placed on the floor of the tank to provide enrichment (mean holding tank temperature [± SD]: cyprinids = 17.9 [± 0.15] °C and anguillid = 13.2 [± 0.89] °C). Due to the silver eel migratory life stage no feeding took place. Cyprinids were fed daily with commercially available flaked food. The tanks were fitted with gravity fed external filters and UV filtration systems. A venturi system on the filter outlets provided aeration to supplement that provided by large capacity air pumps. Fish health and water quality were monitored daily, ensuring consistent conditions were maintained (pH: 7.8–8.4, Ammonia: 0 ppm, Nitrite: 0 ppm, Nitrate: <40 ppm). In line with our standard operating procedures, in the event that any fish should exhibit signs of a disease or associated abnormal behaviour they would be removed and euthanized in accordance with the Schedule 1 of the Animals (Scientific Procedures) Act 1986. However, this was not required, and all fish survived to the planned end of the trials. Water temperature within the tanks was recorded using submersible temperature loggers and validated manually on a daily basis. Experimental trials were terminated if the difference in temperature between the holding and experimental tanks exceeded 2 °C. Each fish was used once only to reduce potential for habituation or learnt behaviours. To adhere to the 3Rs, at the end of the anguillid trials (see *Miller et al., 2021*), subject fish were re-utilised in another study (*Currie et al., 2020*). At the end of the cyprinid trials, subject

**Table 1 Characteristics of waveforms used to test electrosensitivity of cyprinid (grass and common carp) and anguillid (European eel) fish with sample sizes provided.**

| Waveform | Pulse width (ms) | Schematic (represents 1 s time frame) | Species tested and sample size (n) |
|---|---|---|---|
| Single pulse- 2 Hz | 100 | | Grass carp ($n = 15$), Common carp ($n = 15$) and European eel ($n = 17$) |
| Double pulse- 2 Hz | 50 | | Grass carp ($n = 15$), Common carp ($n = 15$) and European eel ($n = 17$) |
| Single pulse- 3 Hz | 33.3 | | Grass ($n = 15$) and Common carp ($n = 15$) |
| Double pulse- 3 Hz | 16.7 | | Grass ($n = 15$) and Common carp ($n = 15$) |

fish were euthanized in accordance with Schedule 1 of the Animals (Scientific Procedures) Act 1986 by an overdose of an anaesthetic followed by confirmation of death by destruction of the brain.

### Experimental procedure

Trials using cyprinids were conducted from 10 August–4 September 2018. Grass carp are thought to have similar activity levels during the day and night (*Mitzner, 1978*), but common carp swimming activity is suggested to be higher at night (*Rahman & Meyer, 2009*), so trials were conducted between 17:00–06:00 to ensure direct comparison between species. Trials using adult European eel were conducted from 2–8 November 2017 between 17:00–02:00 (*Miller et al., 2021*) to replicate conditions experienced during the natural nocturnal downstream migration (*Tesch, 2003*). Ambient light levels during the trials were less than 0.01 lux.

The pulse generator was used to produce four different waveforms: (a) single pulse- 2 Hz, (b) double pulse- 2 Hz, (c) single pulse- 3 Hz and (d) double pulse- 3 Hz (Table 1). For the double pulse- 2 Hz and double pulse- 3 Hz waveforms, the time between the pulses in the set of two (*i.e.,* pulse break) was 50 and 16.7 ms respectively. These waveforms were selected based on ethical considerations as PDC <15 Hz is suggested to reduce injuries in fish (*Sharber et al., 1994*). Comparisons between single and double pulse were performed as previous research has suggested this can elicit differences in behavioural responses (*Bowen et al., 2003*). Furthermore, similar frequencies to those tested in this study have also proved effective at guiding other species (2 Hz: white sturgeon, *Acipenser transmontanus*, *Ostrand et al., 2009*; 3 Hz: Coho salmon, *Oncorhynchus kisutch*, *Raymond, 1956*) and enabled a direct comparison with eel (*Miller et al., 2021*). To generate the desired field strength the input voltage on the pulser was divided by the distance between the electrodes and then verified using a custom-built probe connected to an oscilloscope (Gwinstek GDS-1052-U). Treatment waveforms were alternated across trials in a systematic design to ensure an even spread of treatments.

Prior to the start of each trial, a single individual fish was placed in the experimental area between the mesh screens (Fig. 1) and allowed to settle for 10 min (see *Miller et al., 2021*). This was followed by a 10 s control period (0 V/cm) and a 10 s treatment of 0.05 V/cm and subsequent 10 min recovery. The cycle of 10 s –10 s control-treatment followed by recovery was repeated with increasing field strength in increments of 0.05 V/cm until *tetany* was observed. The response (*no response, twitch, loss of orientation, tetany*) was recorded for each treatment interval.

Experimental tank temperature was measured before (mean temperature start [± SD] °C: cyprinid = 19.4 [± 0.78] °C, anguillid = 13.35 [± 0.61] °C) and after each trial (mean temperature end [± SD] °C: cyprinid = 19.5 [± 0.78] °C, anguillid = 13.44 [± 0.63] °C) and fish were weighed (mean mass [± SD] g: grass carp = 17.4 [± 5.3] g, common carp = 26.3 [± 7.45] g and anguillid = 334.4 [± 94.4] g) and measured when the trial was terminated (mean fork length [± SD] mm: grass carp = 110.3 [± 10.9] mm and common carp = 105.3 [± 16.7] mm; mean total length [± SD] mm: grass carp = 121.0 [± 10.8] mm, common carp = 118.4 [± 10.9] mm and eel = 558.7 [± 52.2] mm).

### Fish physiology

Physiological responses (see *Miller et al., 2021*) (Table 2) were based on experimental observations under the pulse frequencies and widths specified. Due to the involuntary nature of responses, these were classified as physiological. The lowest electric field strength measured that elicited each response was quantified as the threshold strength for that physiological response.

### Statistical analyses

Statistical analyses were conducted using R 3.5.1 (*R Core Team, 2018*). Data was visually inspected for normality before conducting a Shapiro–Wilk test. In the case that the data was non-parametric efforts were made to transform it to achieve normality, and if unsuccessful non-parametric tests were performed. For both grass and common carp, the field strengths for *twitch, loss of orientation* and *tetany* were compared across waveforms (*Miller et al., 2021*) with respect to pulse width and frequency using Kruskal-Wallis tests. Kruskal-Wallis tests were also performed to test for differences in threshold field strengths between cyprinid species (grass and common carp) and families (cyprinid and anguillid). Post-hoc analyses were performed using the Dunn's Test.

## RESULTS

### Threshold field strengths for physiological responses across waveforms (Objective 1)

For grass carp, pulse width influenced the threshold field strengths for *tetany* ($\chi^2(3) = 12.4$, $p = 0.006$) but not *twitch* and *loss of orientation* (*twitch*: $\chi^2(3) = 1.06$, $p = 0.79$, *loss of orientation*: $\chi^2(3) = 2.69$, $p = 0.44$) (Fig. 2A). Post-hoc analyses revealed *tetany* was elicited at a lower field strength in the single pulse-2 Hz waveform treatment than the double pulse-3 Hz (Dunn's Test: $z = -3.21$, $p = 0.008$). Pulse frequency did not affect threshold field strengths for any of the responses (*twitch*: $\chi^2(1) = 0.22$, $p = 0.64$, *loss of orientation*: $\chi^2(1) = 0.15$, $p = 0.7$ and *tetany*: $\chi^2(1) = 1.57$, $p = 0.21$).

**Table 2** Definitions of physiological responses exhibited by grass and common carp and adult European eel in response to electric fields (from *Miller et al., 2021*).

| Metric | Definition |
| --- | --- |
| *No response* | No change or alteration in swimming movements on experiencing an electric pulse |
| *Twitch* | Twitching or jerking movements of the fish body in synchrony with an electric pulse |
| *Loss of orientation* | Loss of vertical body orientation, rapid but uncontrolled swimming behaviour, collision with side walls of test tank |
| *Tetany* | Muscular contraction of entire body and cessation of swimming, fish recover immediately after stimulus removed |

For common carp, threshold field strengths for *twitch*, *loss of orientation* and *tetany* were not affected by pulse width (*twitch*: $\chi^2(3) = 1.80$, $p = 0.62$, *loss of orientation*: $\chi^2(3) = 0.41$, $p = 0.94$, *tetany*: $\chi^2(3) = 4.55$, $p = 0.21$) (Fig. 2B). Pulse frequency also had no effect on threshold field strengths for *twitch* ($\chi^2(1) = 0.19$, $p = 0.67$), *loss of orientation* ($\chi^2(1) = 0.07$, $p = 0.79$) or *tetany* ($\chi^2(1) = 0.05$, $p = 0.82$).

## Threshold field strengths for physiological responses of two cyprinid species (Objective 2)

With the exception that higher field strengths were required to elicit *loss of orientation* for common carp than grass carp under the single pulse- 3 Hz treatment ($\chi^2(1) = 5.51$, $p = 0.02$), there was no difference in thresholds of *twitch*, *loss of orientation* or *tetany* under any of the treatments ($p > 0.05$) (Figs. 3A, 3B, 3C, 3D).

## Threshold field strengths for physiological responses of carp and eel (Objective 3)

As there were no differences in the threshold responses between grass and common carp for the single pulse- 2 Hz and double pulse- 2 Hz (Figs. 3A, 3B), data was aggregated to compare carp (cyprinids) with European eel (anguillid) (*Miller et al., 2021*).

The field strength at which the eel exhibited threshold responses for *twitch* (single pulse- 2 Hz: $\chi^2(1) = 4.34$, $p = 0.04$; double pulse- 2 Hz: $\chi^2(1) = 8.5$, $p = 0.004$), *loss of orientation* (single pulse- 2 Hz: $\chi^2(1) = 27.5$, $p < 0.0001$; double pulse- 2 Hz: $\chi^2(1) = 26.6$, $p < 0.0001$), and *tetany* (single pulse- 2 Hz: $\chi^2(1) = 28.9$, $p < 0.0001$; double pulse- 2 Hz: $\chi^2(1) = 28.1$, $p < 0.0001$) was lower than for carp under both treatments (Figs. 4A, 4B).

## DISCUSSION

To inform the development of selective fish guidance systems using electric fields, this study determined the electrosensitivity to different pulsed direct current waveforms of two cyprinids, grass and common carp, and compared the results with those obtained for an anguillid species of high conservation concern, the European eel (*Miller et al., 2021*). For both cyprinid species the threshold field strengths at which three key physiological responses (*twitch*, *loss of orientation* and *tetany*) were elicited were largely unaffected by the electric field parameters (pulse frequency and width), with one exception; grass carp exhibited

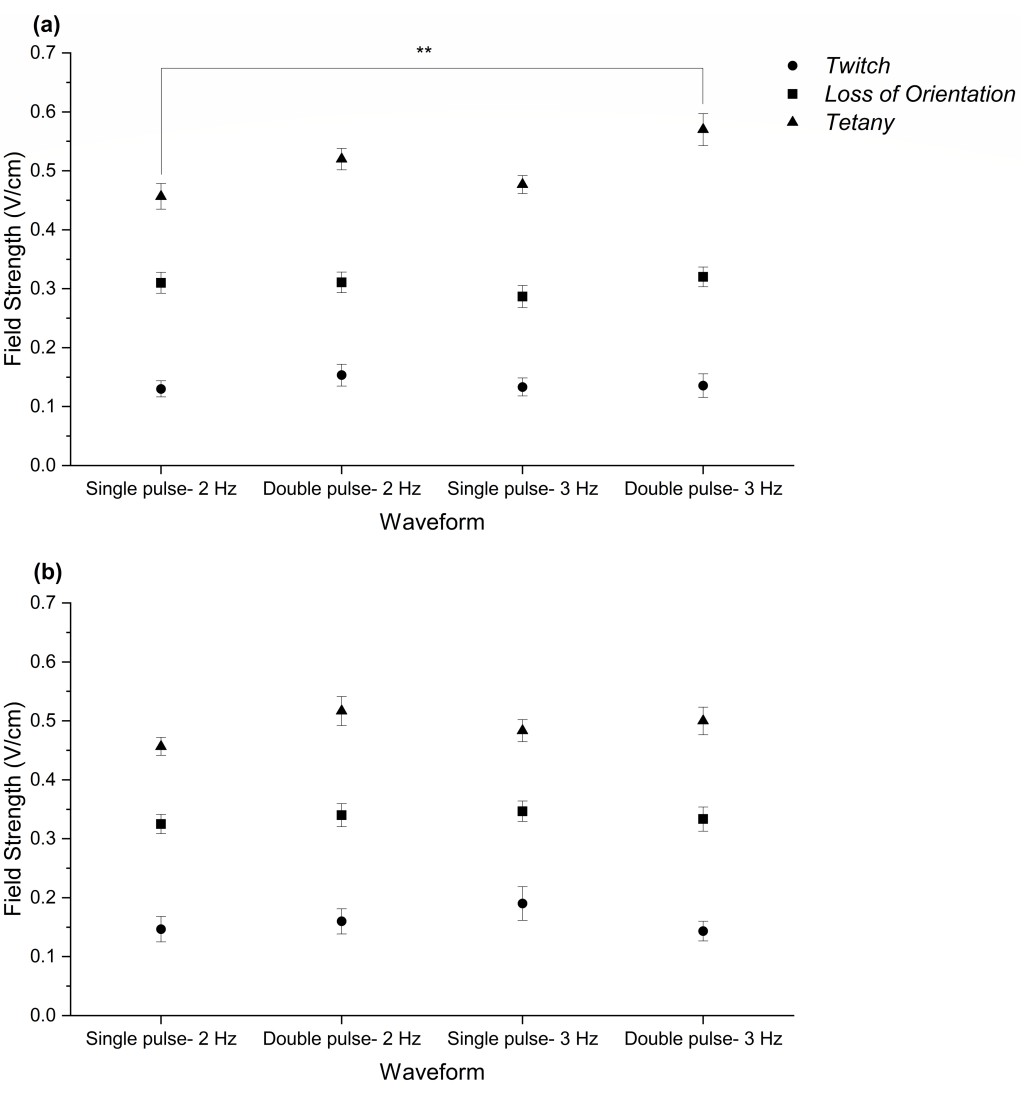

**Figure 2** Mean threshold field strengths (±SE) for three physiological responses; *twitch* (circles), *loss of orientation* (squares) and *tetany* (triangles) exhibited by each cyprinid species. (A) Grass carp and (B) common carp in four waveforms treatments (single pulse- 2 Hz, double pulse- 2 Hz, single pulse- 3 Hz and double pulse- 3 Hz). Note: ** $p < 0.01$.

*tetany* at lower field strengths under the single pulse- 2 Hz than the double pulse- 3 Hz. Electrosensitivity was similar between cyprinids except in the single pulse- 3 Hz waveform treatment where *loss of orientation* occurred at a slightly higher field strength for common carp. The threshold field strengths for all physiological responses were higher for both cyprinid species (*i.e.,* they exhibited lower electrosensitivity) than for adult European eel, presumably because of the latter's longer body length. This study provides the foundations for future research to further develop selective guidance / deterrent systems using electric fields.

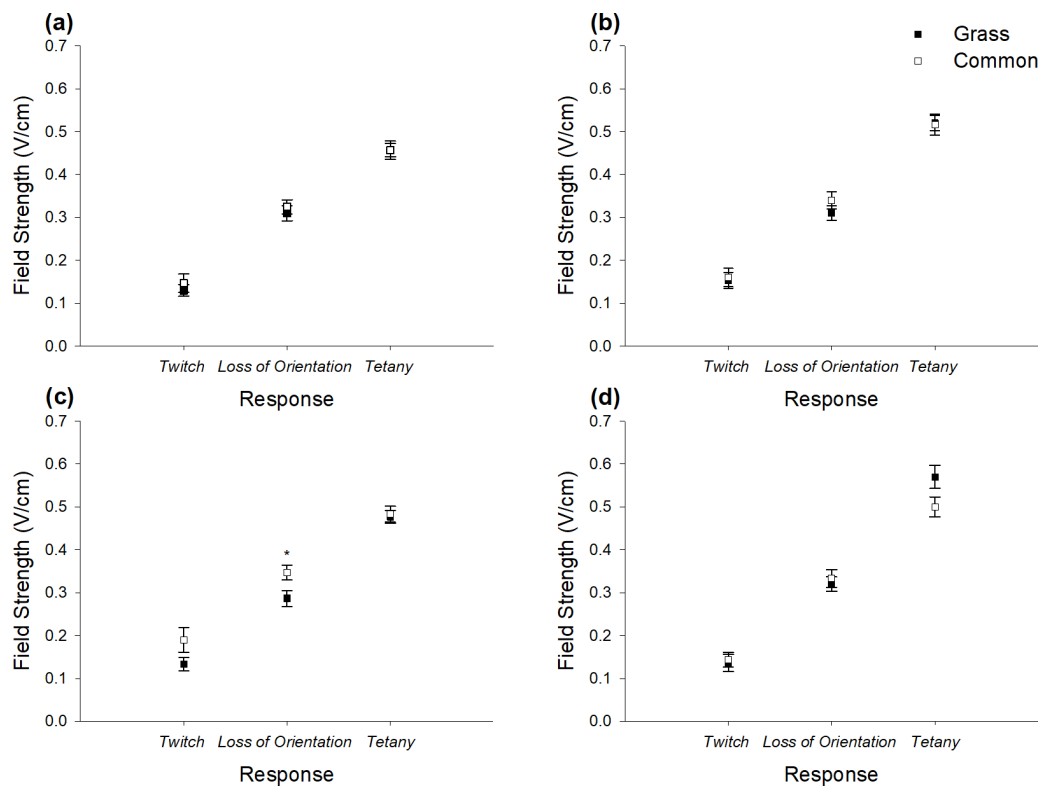

**Figure 3** Mean (±SE) threshold field strengths for both grass and common carp for physiological responses (*twitch, loss of orientation* and *tetany*) across four waveforms tested. (A) single pulse-2 Hz, (B) double pulse-2 Hz, (C) single pulse-3 Hz and (D) double pulse-3 Hz. Note: * $p < 0.05$.

At field strengths higher than 0.4 V/cm both cyprinid species exhibited unvolitional *tetany*. At field strengths above this value effective guidance that requires the modification of a volitional response may be prohibited. Similar field strengths (0.2–0.4 V/cm) are known to be effective at inhibiting common carp movement in the laboratory (*Kim & Mandrak, 2017*). Conversely, in field application higher field strengths (0.79–0.91 V/cm) are employed to control cyprinid movement in the Chicago Sanitary Shipping Canal (*Parker et al., 2015*), with the justification that native species do not migrate through the area (*Moy et al., 2011*) and smaller silver carp are likely to have a lower electrosensitivity due to the voltage gradient measured across their body (anterior to posterior) (*Dolan & Miranda, 2003*; *Parker et al., 2015*). However, care must be taken when comparing results due to differences in the waveform parameters tested and environmental variables such as water conductivity as this will directly influence the power transferred to the fish.

Minimal differences in threshold field strengths of response were observed across waveforms with respect to pulse width and frequency. However, for grass carp, *tetany* occurred at lower field strengths in the single pulse-2 Hz treatment than for the double pulse-3 Hz waveform, presumably due to the longer pulse width in the former (100 ms compared to 16.7 ms). This would support observations related to current applications in which higher peak voltages (power) are needed to immobilise fish when shorter pulse

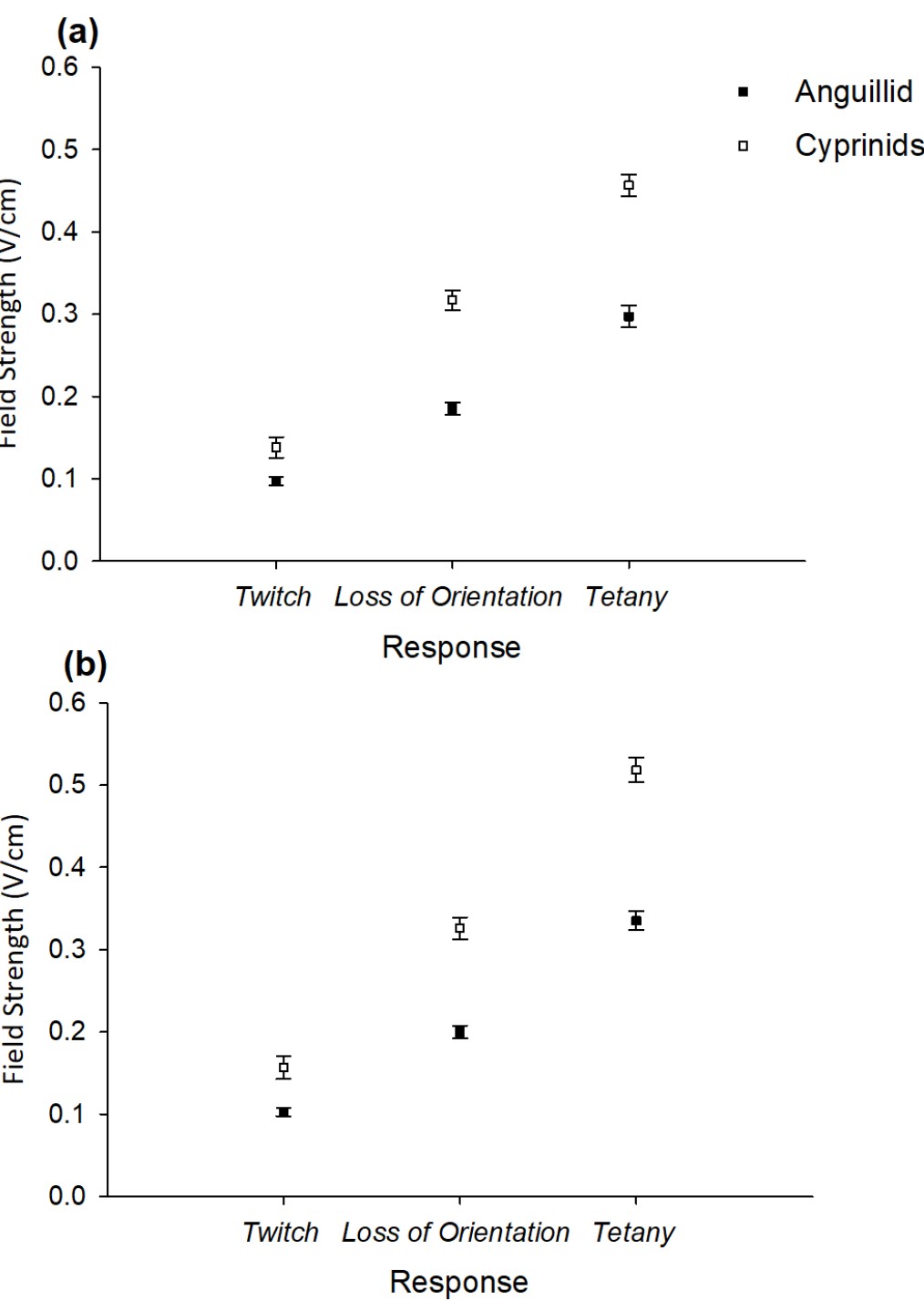

**Figure 4** Mean (±SE) threshold field strengths in the (A) single pulse-2 Hz and (B) double pulse-2 Hz treatments for three physiological responses *twitch*, *loss of orientation* and *tetany* exhibited by carp (cyprinids) and eel (anguillid).

widths are used in electric fishing (*Dolan & Miranda, 2003*). Furthermore, longer pulse widths reduce the probability of juvenile and adult rainbow trout (*Oncorhynchus mykiss*), passing an electric barrier (*Layhee et al., 2016*). Interestingly, pulse width did not have a similar and consistent influence on other responses, suggesting that the most extreme physiological response observed (*tetany*) is most sensitive to differences in pulse width.

Pulse frequency did not influence any of the threshold field strengths for the responses considered for either grass or common carp. The absence of an effect might be explained by the minimal difference in pulse frequency between the waveforms employed (2 *vs.* 3 Hz). We selected relatively low frequencies because: (1) previous observations indicate they are effective at modifying fish movements (*e.g.*, *Mesa & Copeland, 2009* for steelhead, *Oncorhynchus mykiss* and Pacific lamprey, *Entosphenus tridentatus* at 3 Hz and 2 Hz; *Savino et al., 2001* for round Goby, *Neogobius melanostomus*, at 2 Hz); (2) the risk of injury is reduced compared to higher frequency waveforms (*e.g.*, *Culver & Chick, 2015*; *Miranda & Kidwell, 2010*) and (3) low frequency waveforms can be generated using less power, under a fixed millisecond pulse width, and thus reduce economic costs and the carbon footprint compared to other guidance systems (*Beaumont, 2016*).

Both cyprinid species exhibited similar electrosensitivities with no differences in the response to the single pulse-2 Hz, double pulse-2 Hz and double pulse-3 Hz observed. However, the one exception was the *loss of orientation* exhibited by grass carp at lower field strengths than for common carp in the single pulse-3 Hz waveform treatment. Previous work indicates that the threshold field strength for immobilisation is negatively related to the length of the fish (*Briggs et al., 2019*). In this study, grass carp were longer than common carp, but the difference was slight (*e.g.*, 5 mm in mean fork length). It may be possible that *loss of orientation* is more sensitive to differences in body length than the other physiological responses, but further work is needed to confirm this. An alternative explanation is that even among closely related species there is still the possibility for variation in physiological response under specific parameters that could result in slight differences in the effectiveness of different electric field treatments for selective guidance systems.

For all the physiological responses exhibited, lower threshold field strengths were required for adult eel than both cyprinid species (*Miller et al., 2021*). The most likely explanation is the considerably longer body length of the adult eel compared to carp resulting in a greater anterior-posterior voltage differential if parallel to field lines (*Reynolds, 1996*). The difference in electrosensitivity may also reflect physiological differences between the two families.

This study quantified the threshold field strengths at which physiological responses were exhibited for two carp species of considerable conservation concern due to their invasiveness, including in European Rivers (*e.g.*, *Lehtonen, 2002*; *Van der Veer & Nentwig, 2015*), and compared these with previously collected data for European eel, a species that is critically endangered in Europe. Eel have a higher electrosensitivity than carp, while the two carp species largely responded to the electric field treatments in a similar way. The results may have value in informing the development of selective multi-species barriers and guidance devices that could employ electric fields where these species occur in sympatry (*Haubrock et al., 2019*) and negative impacts on eel might be expected. The

differences in electrosensitivity exhibited between families is promising in application to advancing an integrated pest management programme in which there is an interest in the conservation of native species at risk of river infrastructure. Globally, this scenario applies to multiple species where both those that are invasive co-occur with those for which conservation programmes focus on re-establishing river connectivity. This approach will allow trapping of those deemed undesirable while facilitating the free movement of those of high conservation value. This study provides a first step in the design of future selective guidance systems. Field studies should be performed in the future to validate the results obtained and optimise the parameters of the electric fields used to account for site specific characteristics (*e.g.*, water and sediment conductivity).

## ACKNOWLEDGEMENTS

We thank Paul Jacobson (Electric Power Research Institute) for the coordination of this project, Scott Miehls (United States Geological Survey) for the loan of pulser unit, Steve Walk (United States Geological Survey) for the modification of pulser units for experimentation, and Alex Haro (United States Geological Survey) for advice on methodological approach.

### Funding

This study was funded by the Engineering and Physical Sciences Research Council (EPSRC) (Grant Number: EP/L01582X/1) and the Electric Power Research Institute (EPRI). The funders had no role in study design, data collection and analysis, decision to publish, or preparation of the manuscript.

### Grant Disclosures

The following grant information was disclosed by the authors:
The Engineering and Physical Sciences Research Council (EPSRC): EP/L01582X/1.
The Electric Power Research Institute (EPRI).

### Competing Interests

The authors declare there are no competing interests.

### Author Contributions

- Mhairi Miller conceived and designed the experiments, performed the experiments, analyzed the data, prepared figures and/or tables, authored or reviewed drafts of the article, and approved the final draft.
- Suleiman M. Sharkh conceived and designed the experiments, analyzed the data, authored or reviewed drafts of the article, and approved the final draft.
- Paul S. Kemp conceived and designed the experiments, analyzed the data, authored or reviewed drafts of the article, and approved the final draft.

## Animal Ethics

The following information was supplied relating to ethical approvals (i.e., approving body and any reference numbers):

All methods were followed in accordance with the UK national regulatory Animal Welfare Ethical Review Body's (AWERB) guidelines and regulations. This study was approved by the University of Southampton's Ethics and Research Governance Office (Ethics ID 45107 & 30639) and reported in accordance with ARRIVE guidelines.

## Data Availability

The data are available at the University of Southampton repository: Miller, Mhairi Catriona, Sharkh, Suleiman and Kemp, Paul (2023) Dataset to support the publication: Developing electric barriers for selective-fish guidance: Interspecific variation in response to pulsed direct currents. University of Southampton doi: 10.5258/SOTON/D2529 [Dataset].

## Supplemental Information

Supplemental information for this article can be found online at http://dx.doi.org/10.7717/peerj.17962#supplemental-information.

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
