# Peer review of "Laying the foundations for selective-fish guidance using electricity: multi-species response to pulsed direct currents"

_PeerJ, doi:10.7717/peerj.17962_

## Round 0.1 · original submission · Major Revisions

Although both reviewers have noted clear strengths in your manuscript, I invite you to make revisions before it can be accepted for publication. Both reviewers have raised concerns about the difference in length between the two species being compared, as length has a significant effect on electrical deterrence. This should be clarified. There is a need to include a hypothesis guiding the study, and Reviewer 2 has provided a suggestion you can consider. These and the rest of the comments should be considered exhaustively during revision, and details provided on how the comments and suggestions have benefitted the manuscript.

Reviewer 1 ·

Basic reporting

Overall the writing was clear and well organized. The figures and tables were relevant and easily interpreted.

Experimental design

The experimental design is well thought-out. Certainly, the level of detail in the materials and methods are sufficient for others to recreate the study. The only caveat would be that body length of fish was not properly controlled for between the carps and eels (See general comments). All ethical approvals are documented.

Validity of the findings

Fundamentally, the major findings of the study can be summarized in reference to Lines 144-147: 1) the threshold field strength at which three key physiological responses were elicited were largely unaffected by the electric field parameters and minimal differences in responses were observed across waveforms; 2) both cyprinid species exhibited similar electrosensitivities with no interspecific differences; and 3) differences in responses between eel and carps was likely due to longer body length. The major findings of this study (above, using nearly the exact text of the manuscript) do not support the overarching conclusion that interspecific variation is the cause of differences in responses. The data presented seems to suggest an alternate conclusion that there was little intraspecific variation for cyprinids and that body length was the most likely cause of threshold differences between cyprinids and anguillids.

Fundamentally, the data presented is clear and the findings should be of general interest to the scientific and fish management community. However, the authors conclusions seem to be out of alignment with the data (See general comments).

Additional comments

Line 40-42 – This is not supported by the manuscript. The authors fail to provide a compelling site where these species coexist at these life stages nor do the authors acknowledge the substantial difference in length between the carps and eel. Due to the known dependence of fish length on the effect of electrical deterrence and difference in sizes between carp and eel, it remains unclear whether size specific or interspecific variation is the cause for the difference in response.

Line 113 –The terminology “Asian carp” is no longer recommended for reference to these species (See Kočovský et al. 2018).

Kočovský, P.M., Chapman, D.C. and Qian, S., 2018. “Asian carp” is societally and scientifically problematic. Let's replace it. Fisheries, 43(7), pp.311-316.

Line 349 – This factor undermines much of the authors claims of interspecific variation.
Line 357 – See comment above on the use of “Asian carp”.

Line 388-389 – This statement directly contradicts the claim in the abstract that interspecific variation indicates a potential for selectivity when using PDC.

Line 396-400 - While this alternative explanation could explain the slight difference, the data presented overwhelmingly provides far more evidence to the contrary. Only one combination of waveform and response metric resulted in a significant difference.

Line 412-413 – This statement is overgeneralized considering length was identified as the most likely cause for the difference (Line 403-405).

Line 414-424 – While the goal of selective guidance is certainly important, and this work does contribute to the body of science, I challenge the authors to provide an example where juvenile common carp or grass carp need to be selectively controlled in the presence of adult European eel. Such an example would

Reviewer 2 ·

Basic reporting

(1) English used throughout was clear and unambiguous. Provided a few editorial suggestions in the comments included in the marked-up PDF.
(2) Literature was well referenced throughout and provided a few paper suggestions to look at if they were not already on your literature review list. The discussion may be improved a bit by discussing power transfer theory a bit more in the context of comparing voltage gradients that elicited responses in other species and potential differences in water conductivity.
(3) The tables and figures added to the paper and improved my ability to understand and evaluate the experiments. I appreciated the raw data files and that I could see the responses for each fish tested (and their relevant biological data like size).
(4) The study did not seem to have a strong hypothesis-based logic behind it (comparative biology of electroreception in fishes as an explanation of differences in sensitivity). For a hypothetical example, I didn't see a rationale like..."the number of electroreceptors differ among fishes in different families, therefore, while holding length of fish constant, we hypothesize that XXX species will be more sensitive to PDC because of X,Y, and Z differences in sensory system." Instead, the introduction ends with objective statements that are more or less methodological steps (access waveforms, species, and families of fishes to PDC).

Experimental design

(1) Species specific responses to PDC is important for fishery assessment with electrofishers and selective fish guidance with PDC.
(2) The authors should clarify and justify why the species of fish tested differed so much in length (~400 mm?). If the goal of the study was to compare sensitivity across different species, then controlling for length seems like it would be an important aspect of experimental design (see commented PDF for more details)?
(3) Methods were clear and relatively easy to replicate.

Validity of the findings

I worry that the conclusion about species-specific responses to PDC (eel more sensitive than carp) is confounded presently with differences in the length of these species. The authors may be able to conduct more analyses (regressing length of fish and responses at different V/cm) during a revision to justify their conclusion. Alternatively, the authors could just be clearer in the title and abstract that they documented differences in responses between these species and that the differences could be associated with differences in length or species or both. Regardless, PDC may offer opportunities for species- and size-specific sorting of fishes and this paper provides some much-needed quantitative data.

Additional comments

See comments in PDF.

Annotated reviews are not available for download in order to protect the identity of reviewers who chose to remain anonymous.

---

## Round 0.2 · Minor Revisions

Before I can accept your manuscript for publication you need to address the additional comments raised by the reviewer.

Reviewer 1 ·

Basic reporting

The writing is clear and of generally good quality.

Experimental design

The experimental design is adequate, but the rationale behind selection of juvenile common and grass carp and adult European eel is lacking. Further comments are provided in the additional comments section.

Validity of the findings

No comment

Additional comments

Line by line comments:

Line 38-43: The abstract does not mention the primary finding that body length was the determining factor for variation in electrosensitivity between carps and eel. This is an underlining explanation for the most striking species difference and should not be avoided.

Line 45: The authors response indicates that the responses observed in this study were primarily physiological in nature, not behavioral. Yet, “behavioral barrier” is listed as a keyword. Given the other keywords, I recommend simply removing it from the list or replacing it with “deterrents”.

Line 142-146: The authors still provide a limited explanation for why these specific species and life stages were selected for the study. Given the significant size difference between eel and both carp species, I would expect the authors have a given site / situation in mind where young carps and adult eel coexist and would need to be guided/blocked selectively. The author’s response, citing a study that proposes to reintroduce eel (not actually coexisting) to control AIS (including common carp) is insufficient. Are there any management level recommendations for this comparison? Otherwise, the primary finding that larger eel are more sensitive small carp is just not novel and seems like a rather random grouping of fish and life stages.

Line 433: Similar to the previous comment, the authors need to do better at describing how or why a system may be needed to separate adult eel from young carps. There are numerous examples in the literature that describe the life histories of all three species. In interconnected lake systems in midwestern USA, common carp tend to spawn in upstream reaches and the juvenile carp swim downstream to return to larger lakes. I presume there are situations where similar behaviors are seen in Europe as well. Given that European eel migrate downstream for spawning, it would not be too onerous to hypothesize a situation where separation of these species may be needed. Without some thoughtful explanation, it is hard to envision what utility this data has. Given the example in the text, my first question would be what happens when larger carp are present? Will they be expected to be trapped?

---

## Round 0.3 · accepted · Accept

The authors have made all the requested corrections, and the manuscript is acceptable for publication.